# Pharmacokinetics, Dose-Proportionality, and Tolerability of Intravenous Tanespimycin (17-AAG) in Single and Multiple Doses in Dogs: A Potential Novel Treatment for Canine Visceral Leishmaniasis

**DOI:** 10.3390/ph17060767

**Published:** 2024-06-11

**Authors:** Marcos Ferrante, Bruna Martins Macedo Leite, Lívia Brito Coelho Fontes, Alice Santos Moreira, Élder Muller Nascimento de Almeida, Claudia Ida Brodskyn, Isadora dos Santos Lima, Washington Luís Conrado dos Santos, Luciano Vasconcellos Pacheco, Vagner Cardoso da Silva, Jeancarlo Pereira dos Anjos, Lílian Lefol Nani Guarieiro, Fabiana Landoni, Juliana P. B. de Menezes, Deborah Bittencourt Mothé Fraga, Aníbal de Freitas Santos Júnior, Patrícia Sampaio Tavares Veras

**Affiliations:** 1Laboratory of Physiology and Pharmacology, Department of Veterinary Medicine, Federal University of Lavras, Lavras 37200-000, Minas Gerais, Brazil; marcos.ferrante@ufla.br; 2Laboratory of Host-Parasite Interaction and Epidemiology, Gonçalo Moniz Institute, Fiocruz-Bahia, Salvador 40296-710, Bahia, Brazil; brunamml@yahoo.com.br (B.M.M.L.); livia.vetcoelho@gmail.com (L.B.C.F.); alice_moreira2@hotmail.com (A.S.M.); elder.muller@gmail.com (É.M.N.d.A.); claudia.brodskyn@fiocruz.br (C.I.B.); juliana.fullam@fiocruz.br (J.P.B.d.M.); deborah.fraga@fiocruz.br (D.B.M.F.); 3Laboratory of Structural and Molecular Pathology, Gonçalo Moniz Institute, Fiocruz-Bahia, Salvador 40296-710, Bahia, Brazil; isadoraslima@hotmail.com (I.d.S.L.); washington.santos@fiocruz.br (W.L.C.d.S.); 4Department of Pathology and Forensic Medicine, Bahia Medical School, Federal University of Bahia, Salvador 40110-906, Bahia, Brazil; 5Department of Life Sciences, State University of Bahia, Salvador 41150-000, Bahia, Brazil; lucianofcd@hotmail.com (L.V.P.); farmoncovagner@gmail.com (V.C.d.S.); afjunior@uneb.br (A.d.F.S.J.); 6Integrated Campus of Manufacturing and Technology, SENAI CIMATEC University Center, Salvador 41650-010, Bahia, Brazil; jeancarlo.anjos@fieb.org.br (J.P.d.A.); lilian.guarieiro@fieb.org.br (L.L.N.G.); 7Department of Pharmacology, Faculty of Veterinary Science, National University of La Plata, Buenos Aires 1900, Argentina; landoni@fcv.unlp.edu.ar; 8Department of Preventive Veterinary Medicine and Animal Production, School of Veterinary Medicine and Animal Science, Federal University of Bahia, Salvador 40170-110, Bahia, Brazil; 9National Institute of Science and Technology of Tropical Diseases (INCT-DT), National Council for Scientific Research and Development (CNPq)

**Keywords:** toxicity, pharmacokinetics, dogs, canine visceral leishmaniasis, Tanespimycin (17-AAG), dose-escalation protocol

## Abstract

In the New World, dogs are considered the main reservoir of visceral leishmaniasis (VL). Due to inefficacies in existing treatments and the lack of an efficient vaccine, dog culling is one of the main strategies used to control disease, making the development of new therapeutic interventions mandatory. We previously showed that Tanespimycin (17-AAG), a Hsp90 inhibitor, demonstrated potential for use in leishmaniasis treatment. The present study aimed to test the safety of 17-AAG in dogs by evaluating plasma pharmacokinetics, dose-proportionality, and the tolerability of 17-AAG in response to a dose-escalation protocol and multiple administrations at a single dose in healthy dogs. Two protocols were used: Study A: four dogs received variable intravenous (IV) doses (50, 100, 150, 200, or 250 mg/m^2^) of 17-AAG or a placebo (*n* = 4/dose level), using a cross-over design with a 7-day “wash-out” period; Study B: nine dogs received three IV doses of 150 mg/m^2^ of 17-AAG administered at 48 h intervals. 17-AAG concentrations were determined by a validated high-performance liquid chromatographic (HPLC) method: linearity (R^2^ = 0.9964), intra-day precision with a coefficient of variation (CV) ≤ 8%, inter-day precision (CV ≤ 20%), and detection and quantification limits of 12.5 and 25 ng/mL, respectively. In Study A, 17-AAG was generally well tolerated. However, increased levels of liver enzymes–alanine aminotransferase (ALT), aspartate aminotransferase (AST), and gamma-glutamyl transferase (GGT)–and bloody diarrhea were observed in all four dogs receiving the highest dosage of 250 mg/m^2^. After single doses of 17-AAG (50–250 mg/m^2^), maximum plasma concentrations (Cmax) ranged between 1405 ± 686 and 9439 ± 991 ng/mL, and the area under the curve (AUC) plotting plasma concentration against time ranged between 1483 ± 694 and 11,902 ± 1962 AUC 0–8 h μg/mL × h, respectively. Cmax and AUC parameters were dose-proportionate between the 50 and 200 mg/m^2^ doses. Regarding Study B, 17-AAG was found to be well tolerated at multiple doses of 150 mg/m^2^. Increased levels of liver enzymes–ALT (28.57 ± 4.29 to 173.33 ± 49.56 U/L), AST (27.85 ± 3.80 to 248.20 ± 85.80 U/L), and GGT (1.60 ± 0.06 to 12.70 ± 0.50 U/L)–and bloody diarrhea were observed in only 3/9 of these dogs. After the administration of multiple doses, Cmax and AUC 0–48 h were 5254 ± 2784 μg/mL and 6850 ± 469 μg/mL × h in plasma and 736 ± 294 μg/mL and 7382 ± 1357 μg/mL × h in tissue transudate, respectively. In conclusion, our results demonstrate the potential of 17-AAG in the treatment of CVL, using a regimen of three doses at 150 mg/m^2^, since it presents the maintenance of high concentrations in subcutaneous interstitial fluid, low toxicity, and reversible hepatotoxicity.

## 1. Introduction

The leishmaniases are zoonoses of significant public health importance widely distributed around the world. Visceral leishmaniasis (VL) is endemic in 68 countries (the WHO, 2018), and Brazil has one of the highest rates of incidence. Mainly caused by *Leishmania infantum*, VL is transmitted to mammals through the bite of the insect vector *Lutzomyia longipalpis*. Dogs are considered the main reservoir in domestic and peridomestic environments, contributing to the maintenance of the VL cycle [1,2,3]. Canine VL (CVL) is a chronic systemic disease of significance to human and veterinary health [4,5]. Infected dogs can present a range of clinical forms, from subclinical infection to severely debilitating clinical conditions [6,7,8,9], similar to human VL [10,11,12,13,14].

Treating dogs with VL presents a significant challenge in the veterinary clinical routine due to parasites’ persistence. Frequently, dogs treated with currently available medications do not achieve a parasitological cure despite a lessening in the severity of clinical signs or apparent cure in addition to decreases in the titers of antibodies and parasitic load in lymphatic tissues [15,16]. Among the drugs indicated for CVL treatment, the most common are pentavalent antimonials and conventional or liposomal formulations of amphotericin B, allopurinol, pentamidine, miltefosine, and immunotherapy [16,17,18,19,20].

Given the high rates of therapeutic failure to CVL treatment, high toxicity that may cause death, parasite resistance, and high cost, the need to discover new chemotherapeutic agents to maximize therapeutic effects and reduce side effects is paramount. This need has led to the study of novel possible targets, such as heat shock protein inhibitors 90 (Hsp90). Hsp90 inhibitors, including those belonging to the family of benzoquinone ansamycins, e.g., geldanamycin (GA), 17-(allylamino)-17-demethoxygeldanamycin (17-AAG) or Tanespimycin, and 17-dimethylaminoethylamino-17-demetoxigeldanamycin (17-DMAG), have been extensively investigated as anti-cancer drugs [21,22,23,24]. Among these inhibitors, 17-AAG was shown to curb the advance of carcinogenesis in preclinical studies both as a single agent and in association with other anti-cancer agents against several types of cancer [25,26,27]. It is also the first Hsp90 inhibitor used in human clinical trials for cancer treatment due to lower toxicity than GA and more significant affinity to the Hsp90 ATP binding site, exerting a more significant inhibitory effect [28]. Hsp90 inhibitors have also been used to control parasitic infections due to the essential role of Hsp90 in maintaining the parasite cycle and adaptation to stress [29,30]. Regarding *Leishmania* infection, the potential antileishmanial properties of 17-AAG have been demonstrated against *Leishmania amazonensis* and *Leishmania braziliensis* [31,32,33].

Clinical pharmacokinetics is focused on optimizing pharmacological treatments to subsequently achieve maximum therapeutic efficacy with minimum adverse effects and physiological, biochemical, and cellular alterations [34,35]. Even the most promising pharmacological therapies will fail in clinical studies if the drug cannot reach target organs sufficiently to exert a therapeutic effect [36]. Compartmentalized and non-compartmentalized models are currently being used to estimate pharmacokinetic parameters [37,38]. The compartmentalized modeling describes a drug’s fate by dividing the whole body into one or more pharmacokinetically distinct homogenous compartments, including various tissues and organs in chemical equilibrium [39,40]. On the other hand, direct non-compartmentalized modeling calculates the area under the curve (AUC) based on drug concentrations over time [41]. This modeling adequately describes the dissemination over time of newly introduced compounds in animals and is most often used in preclinical research [38,42]. The use of sensitive and robust analytical techniques, such as high-performance liquid chromatographic (HPLC), has contributed to the advancement of clinical pharmacokinetics, especially in the development of methods for the determination of Tanespimycin (17-AAG) in biological fluids [43,44,45].

Human evaluations of the pharmacokinetics have shown that 17-amino-geldanamycin (17-AG) is the active metabolite of 17-AAG [46,47]. The half-life of 17-AAG is 3–6 h, while 17-AG is 6.2–7.6 h. The urinary excretion of 17-AAG in humans has been estimated to be less than 10% of a specific dose, which becomes rapidly eliminated by the hepatobiliary system [48,49]. For the later evaluation of 17-AAG antileishmanial efficacy in dogs, preclinical studies were carried out in healthy dogs. We aimed to determine relevant pharmacokinetics parameters. Subsequently, we evaluated plasma pharmacokinetics and toxicity, estimated dose proportionality, and the tolerability of 17-AAG in response to single and multiple intravenous (IV) doses. 

## 2. Results

### 2.1. Study A—Dose-Escalation Protocol

#### 2.1.1. Evaluation of 17-AAG Pharmacokinetics in Plasma

In Study A, concentrations of 17-AAG were estimated in plasma samples from four dogs following the administration of each dose concentration (50, 100, 150, 200, and 250 mg/m^2^). The AUC was calculated for each 17-AAG plasma concentration–time point to estimate the pharmacokinetic parameters, which are summarized in Table 1. Significant differences in AUC and Cmax were observed when comparing doses of 50, 100, 150, and 200 mg/m^2^, as well as between 50, 100, and 250 mg/m^2^. However, no differences were found between doses of 150 or 200 mg/m^2^ and 250 mg/m^2^. We only found differences between half-life (t½) values between doses of 150 and 200 mg/m^2^. Clearance of 17-AAG from plasma was, on average, 0.026 ± 0.010 L/h/m^2^, with a coefficient of variation (CV) of 36%, while the CV between animals ranged from 1% to 20% and was not statistically different among doses. Similarly, no significant differences were observed among the various doses for distribution volume (DV) (Table 1). Figure 1 and Table 1, respectively, illustrate that both AUC (measured over time) and Cmax increased proportionally with administered doses of 17-AAG, showing a linear correlation between doses ranging from 50 to 200 mg/m^2^. However, no correlation was observed between plasma concentration and the dose of 250 mg/m^2^ (Figure 2). Also, we observed a rapid decline of 17-AAG in plasma regardless of the dose applied (Figure 2). 

#### 2.1.2. Assessments of 17-AAG Tolerability

The appearance of biochemical, gastrointestinal, and other system alterations was shown to be dose-dependent, occurring more frequently in dogs treated with higher doses (200 and 250 mg/m^2^) than in lower doses (Table 2). The administration of 17-AAG in concentrations of 50 to 150 mg/m^2^ was considered safe and even associated with reversible undesirable effects. No alterations were observed during and after the intravenous administration of the placebo. 

In Study A, all four animals presented nausea (following the parameters described by Kenward et al., 2014 [50]) during the administration of 200 mg/m^2^ and 250 mg/m^2^ of the compound (Table 2). After drug administration, diarrhea with pasty feces was noted in 25% of the animals at doses of 150, 200, and 250 mg/m^2^; bloody diarrhea was present in 75% of the dogs at 200 mg/m^2^ and in 100% at 250 mg/m^2^; and vomiting occurred in 25% at both 200 and 250 mg/m^2^. Other side effects observed in dogs following the administration of 17-AAG were pruritus, erythema, and hyperthermia (Table 2).

### 2.2. Study B–Multiple Administrations at a Single Dose of 150 mg/m^2^

#### 2.2.1. Analysis of the Pharmacokinetics of 17-AAG in Plasma and Subcutaneous Interstitial Fluid

The plasma concentrations of 17-AAG were observed to decline rapidly and reached concentrations close to 100 ng/mL in the subcutaneous interstitial fluid at all times evaluated. Multiple administrations of 17-AAG did not lead to any detectable accumulations in plasma in evaluations conducted at 48 h intervals (Figure 2 and Figure 3). Regarding the pharmacokinetic differences among dogs given three doses of 150 mg/m^2^ of 17-AAG, a significant difference was observed only in the t**½** between the first and third administrations. No changes in Cmax values were observed in either plasma or interstitial fluid at any of the assessments undertaken after each 150 mg/m^2^ dose. Significant differences in AUC were observed when comparing the time intervals of 0–8 h with 0–24 h and 0–6 h with 0–24 h. No significant differences were found among other intervals or across different administrations (Table 3 and Table 4).

#### 2.2.2. Assessments of 17-AAG Safety and Toxicity

During the administration of the first and third doses of 150 mg/m^2^ of 17-AAG, only one animal showed a skin reaction with temporarily erupted hairs. On the second day after administering the second dose, two out of nine animals presented with pasty feces. On the third day, three had diarrhea with liquid feces, one showed bloody diarrhea, and two showed pasty feces out of nine dogs. On the fourth day, seven out of nine animals had pale feces, and one animal had bloody diarrhea (Table 5). When considering the hematological parameters and plasma biochemistry in animals after the administration of 17-AAG, only the total protein showed a significant difference among dogs in relation to drug concentration. Dogs receiving higher doses exhibited elevated levels of total protein in their serum. Plasma biochemistry analyses revealed that the enzymes, alanine aminotransferase (ALT) (28.57 ± 4.29 to 173.33 ± 49.56 U/L), aspartate aminotransferase (AST) (27.85 ± 3.80 to 248.20 ± 85.80 U/L), and gamma-glutamyl transferase (GGT) (1.60 ± 0.06 to 12.70 ± 0.50 U/L) were shown to be elevated in treated dogs, but these differences were revealed not to be statically distinct and were reversible after drug washout.

When analyzing the frequency of side effects in dogs receiving varying doses of 17-AAG, increases in AST were noted in the plasma of dogs treated with 100 mg/m^2^ (50% of animals), 150 mg/m^2^ (50%), 200 mg/m^2^ (100%), and 250 mg/m^2^ (100%). However, these elevations were reversible. In animals where multiple doses of 17-AAG at 150 mg/m^2^ were administered, elevated levels of ALT and AST were observed in only 33% of the dogs (three out of nine). By the fourth day, elevated transaminase levels persisted in only one animal (Table 5).

#### 2.2.3. Histopathology Analysis of Different Tissues of Dogs after Treatment

Histopathological changes were found in the kidney, liver, spleen, small intestine, and large intestine (Table 6). In the lung and heart, no histopathological alterations were observed in any of the treated animals. In the kidney, in 100% (9/9) of the animals, vacuolar degeneration of the renal tubular epithelium was observed (Figure 4A). In the liver, 100% (9/9) of the animals presented with the mobilization of Kupffer cells (Figure 4B). In the spleen, 100% (9/9) of the animals showed congestion, immunoblastic hyperplasia, and spleen type II and III disorganizations [51] (Figure 4C). In the small intestine, areas of calcifications within the intestinal mucosa were found in 22% (2/9) of the animals. In the thick intestine, colitis was detected in 55% (5/9) (Figure 5A,B) of the animals, and 11% (1/9) of the dogs presented areas of calcifications within the mucosa of the thick intestine.

## 3. Discussion 

The present study evaluated the pharmacokinetics and safety of intravenously administered Tanespimycin (17-AAG) in healthy dogs to assess its therapeutic potential for CVL. Pharmacokinetics analysis detected a peak in plasma concentrations in the first minutes following the intravenous administration of 17-AAG in dogs, with a rapid drop (especially during the first four hours) observed within 10 h. In comparison to plasma, in subcutaneous tissue, we found a lower concentration of 17-AAG, indicating a higher degree of tissue distribution. Egorin et al. [52] also observed a rapid decline in plasma concentrations and a wide distribution of 17-AAG in all murine tissues evaluated after intravenous administration. 17-AAG was detectable for up to 8 h after injection, mostly in the lungs, spleen, and liver, probably due to the low hydrosolubility of the compound. Tanespimycin has a molecular weight of 585.7 g/mol and is soluble in Dimethyl Sulfoxide (DMSO) at 100 mM and ethanol at 10 mM, according to PubChem [53]. Its aqueous solubility is limited, approximately 0.01 mg/mL, and it precipitates when diluted with aqueous liquids at almost any concentration. Consequently, creating a therapeutically suitable formulation of 17-AAG poses significant challenges. However, in this study, we developed a reproducible, sterile, and stable nanodispersed lipid-containing formulation suitable for IV administration, using drug concentration, volume of organic solvent, and Eplerenone (EPL) concentration as critical parameters to prepare an appropriate formulation.

The determination of dose-proportionality is an essential part of the drug development process. It may provide early indications of non-linear pharmacokinetics and identify subpopulations with divergent clearances [54]. AUC and Cmax values indicated proper dose proportionality for 17-AAG between 50 and 200 mg/m^2^ (Figure 1B,D and Table 1).

Several previous studies in mice and humans reported positive correlations between 17-AAG dosage and Cmax and AUC [48,49,52,55]. The observed linearity between AUC and Cmax at doses ranging from 50 to 200 mg/m^2^ enables pharmacokinetics to simulate plasma concentrations of 17-AAG after applying doses within this range and in repeated administration-desired time intervals [54]. In contrast, the application of higher doses resulted in a linearity loss (Figure 1A,C and Table 1). Several reasons could cause the lower AUC values found for a 250 mg/m^2^ dosage: (i) the increased metabolic activity of animals with a greater capacity of drug elimination; (ii) saturation in plasma protein binding, leading to a higher proportion of free drug and, therefore, a more homogeneous distribution and a higher elimination; or (iii) the saturation of transport systems, leading to changes in distribution and excretion processes [56,57]. However, considering that Cmax also decreased at a dosage of 250 mg/m^2^, we speculate that the loss of linearity mainly occurred due to the saturation of the compound in binding to proteins and the transport system or due to the rapid distribution to tissues [52]. 

*L. infantum* is an intracellular parasite with a wide distribution throughout the body. This pathogen can be found in tissues such as the spleen and bone marrow and less-irrigated tissues, such as the lymph nodes and dermis [58]. The tissue cage model allows for the creation of a compartment in peripheral feline and canine tissues that emulates what occurs in diseased animals, thus enabling the characterization of a pharmacokinetic profile in these compartments in association with plasma profiles [59,60,61,62]. Therefore, pharmacokinetic (PK) and pharmacokinetic–pharmacodynamic (PK/PD) simulations can estimate the pharmacological effect in vivo using data from the pharmacokinetic profile in plasma and peripheral compartments where the pharmacological target is located [63,64,65]. This approach is widely used in antibiotic efficacy studies [66,67,68]. It has also been applied in developing an antileishmanial via PK/PD modeling of miltefosine [69,70], which allows for therapeutic dose adjustments to be made using pharmacokinetic profiles [71,72]. The present study attempted to establish the pharmacokinetic profile of 17-AAG not only in plasma but also in subcutaneous tissue cages. Unfortunately, the literature contains no instances of a PK/PD model of *L. infantum* infection using 17-AAG that would allow us to estimate its therapeutic effect against CVL. However, a previous study determined the IC_50_ of 98 ng/mL (169.1 nM) of 17-AAG against *L. infantum* promastigotes in vitro [33]. Concentrations in tissue cages were found to be around 100 ng/mL at 48 h intervals in Study B (Figure 3). This finding suggests that 150 mg/m^2^ intravenously could exert a potential antileishmanial effect in vivo. In future work, to better characterize the pharmacokinetic profile of 17-AAG, it should be advantageous to provide more quantitative data, involving PK/PD analysis and the presentation of the PK/PD relationship using a protein-adjusted IC_50_.

Biochemical, gastrointestinal, and other system alterations associated with the administration of 17-AAG were found to be dose-dependent, with minimal side and adverse effects observed at doses ranging from 50 to 150 mg/m^2^, leading us to consider these concentrations safe for use in dogs. Reversibility in the elevated levels of ALT and AST observed indicates an insignificant potential for cumulative hepatotoxicity. However, hepatotoxicity has been reported to be a primary effect of 17-AAG in humans and dogs, as previous studies found elevated levels of hepatic transaminases and serum bilirubin associated with drug administration [73,74,75,76]. In addition to increased ALT and AST, nausea and diarrhea were also reported in humans treated with 17-AAG at doses ranging from 150 to 340 mg/m^2^ with different treatment regimens involving daily administrations, as well as up to 72 h intervals between doses [49,55,77,78]. Some authors have suggested that a twice-week treatment regimen, including intervals between treatment cycles, could reduce hepatotoxicity [23,77,78,79]. 

After establishing a maximum tolerable dose of 150 mg/m^2^ of 17-AAG in the dogs in Study A, a second study (Study B) was conducted using three doses of 150 mg/m^2^ with 48 h intervals between administrations. Some of the biochemical, gastrointestinal, and other systemic manifestations seen in dogs that received a single dose of 150 mg/m^2^ in Study A were also observed in Study B, namely, skin hypersensitivity and diarrhea. In Study B, only 3/9 dogs presented elevated levels of ALT (28.57 ± 4.29 to 173.33 ± 49.56 U/L), AST (27.85 ± 3.80 to 248.20 ± 85.80 U/L), and GGT (1.60 ± 0.06 to 12.70 ± 0.50 U/L). Yet, alterations were not seen in the blood counts, which stands in contrast to previous reports detailing the presence of anemia and thrombocytopenia in humans treated with variable doses of 17-AAG under various treatment regimens [49,55,77]. 

A previous study reported that the use of Cremophor as a vehicle caused hypersensitivity and anaphylaxis in dogs [80]. Herein, no skin reactions were evidenced in animals that received Cremophor alone. However, 1/9 dogs systemically treated with 17-AAG diluted in Cremophor did present skin hypersensitivity, suggesting that Cremophor in association with 17-AAG could be responsible for this type of reaction. It is worth noting that the hypersensitivity induced by Cremophor can be easily avoided via antihistamine administration [81]. DMSO seems not to be a suitable alternative diluent for 17-AAG since it causes irritation, inflammation, and vascular thrombosis [80]. Toxicity assessments indicated the safety of repeated intravenous administrations of 17-AAG at doses of 150 mg/m^2^. Histopathological changes were mainly observable in the kidney, liver, spleen, and small and large intestines. A previous report described similar alterations in dogs and rodents following the intraperitoneal administration of 17-AAG [76,82,83]. 

One limitation of this study in assessing the safety and tolerability of 17-AAG is its predominant use of female dogs. Ethical considerations significantly constrain the enrollment of healthy dogs, limiting our capacity to recruit new animals for the research. Consequently, the study was conducted with the dogs that were initially recruited. Additionally, it is important to note that the literature lacks information on sex-dependent differences in the pharmacokinetics (PK) of 17-AAG treatment.

Models involving rats and dogs treated with 17-AAG indicated that the liver is a target organ for mild to moderate lesions, leading to increases in AST, ALT, alkaline phosphatase, and GGT in dogs [76,83]. Although it is a less toxic derivative of geldanamycin, 17-AAG still causes serious toxic effects, including hepatotoxicity. In a phase I study conducted by Ramanathan et al. [55], dose-limiting toxicities, including hepatotoxicity, were observed, along with side effects, such as vomiting, nausea, anemia, and myalgias. In addition, histopathological analysis indicated Kupffer cell mobilization in the livers of all nine animals treated with three doses of 17-AAG at 150 mg/m^2^. We suggest that these alterations likely occurred due to an enhancement in liver metabolism secondary to increased cytochrome P450 activity [52,84]. Amin et al. [85] evaluated the toxicity of 17-AAG in canine liver fragments incubated in a medium containing 17-AAG at concentrations ranging from 0.1 to 5 μM. They found that the inhibition of epithelial cell proliferation occurs in a concentration and time-dependent manner and altered levels of AST, ALT, ALP, and GGT in treated fragments. The mechanism related to liver toxicity caused by geldanamycin and its derivatives appears to depend on the benzoquinone moiety of these compounds [86]. However, the precise mechanisms underlying 17-AAG’s hepatotoxicity require further investigation. Samuni et al. [87] suggested that the toxicity against rat primary hepatocytes induced by these Hsp90 inhibitors could be attributed to the generation of reactive oxygen species, such as superoxide.

Among drugs commonly used as first-line treatments for VL, hepatotoxicity has been reported for pentavalent antimonials and the liposomal formulation of amphotericin B (L-AMB). Like 17-AAG, antimonials cause hepatotoxicity along with other severe side effects, including cardiotoxicity and pancreatitis, and symptoms like vomiting, nausea, anorexia, myalgia, and abdominal pain [88]. Kato et al. [89] described that the mechanism underlying hepatotoxicity might involve the accumulation of residual Sb(III) from meglumine antimoniate and that co-treatment with ascorbic acid could reduce hepatic alterations caused by antimonials in infected mice. Their findings support strategies to reduce liver toxicity associated with antimonial therapy in humans using pentavalent antimonials with minimal Sb(III) residue supplemented with ascorbic acid. Hepatotoxicity from treatment with L-AMB was observed in 21% of patients in a retrospective study, potentially related to the compound’s high affinity for binding to biological membranes and lipoproteins, leading to the accumulation of AMB in the liver. This accumulation may elevate transaminase or bilirubin levels, potentially causing organ failure [90]. In contrast to the other drugs in use, the oral treatment of leishmaniasis with miltefosine, which is generally well tolerated in a monotherapy regimen, primarily causes mild gastrointestinal issues, and there is no reported occurrence of hepatotoxicity in either animal models or humans. Similar to the effects seen with 17-AAG in dogs, miltefosine can cause mild-to-moderate elevations in transaminase levels, which typically return to normal upon dosage reduction [91]. Like Glaze et al. [82], the present report found congestion and atrophy in spleen lymphoid tissue and inflammatory lesions in the small and large intestines, in all nine dogs that received three doses of 150 mg/m^2^ during the Study B protocol. While renal toxicity had only been previously described in mice [83], this is the first report that described vacuolar degeneration in the tubular epithelium in the kidneys of all nine dogs from Study B.

The present results demonstrate that doses of 150 mg/m^2^ of 17-AAG lead to moderate, dose-dependent, and reversible liver toxicity, corroborating clinical trials in humans, which indicate the safety of this compound [49,86]. As explained above,17-AAG administration has challenges, since the compound contains a benzoquinone moiety that undergoes reductive metabolism and detoxification by nicotinamide adenine dinucleotide phosphate (NADPH): quinone oxidoreductase (NQ01) (also called DT-diaphorase) before it acts against Hsp90, which is potentially hepatotoxic [92,93]. Another limitation is its low solubility that was surmounted when the 17-desmethoxy-17-N,N-dimethylaminoethylaminogeldanamycin (17-DMAG), a water-soluble derivative of 17-AAG, was discovered. Importantly, the introduction of the ionizable amino group provided the much-needed improvement in water solubility and oral bioavailability and equal, if not greater, antitumor activity than 17-AAG [94]. However, intravenous daily doses of 17-DMAG at 16 mg/m^2^ proved fatal. Toxicity was also observed in response to a daily intravenous dose of 8 mg/m^2^ or 16 mg/m^2^ of 17-DMAG orally, which negates the possibility of using this drug as an antileishmanial in dogs [82].

The major obstacle to the delivery of 17-AAG related to its limited aqueous solubility (ca. 0.01 mg/mL) can encourage studies of special formulations for better solubilization and delivery of the drug by various administration routes, including orally. Although oral administration seems to be a viable route for treatment, given that leishmaniasis is a systemic disease affecting multiple organs, the majority of the literature involving 17-AAG, in both animal models and human studies, predominantly uses intravenous administration [55,95,96,97,98]. According to the National Institute of Cancer’s guide “Turning Molecules into Medicines for Public Health https://dtp.cancer.gov/timeline/posters/AGG_Geldamycin.pdf” (accessed on 16 May 2024), mice with MCF-7 breast tumors treated with various doses of 17-AAG exhibited a maximum plasma half-life of 4.4 h following a 40 mg/kg intravenous dose, yet the drug was undetectable after oral administration. Biopharmaceutical strategies can be useful in this process, such as the incorporation of 17-AAG into micelles with special polymers, liposomes, and other delivery systems on nanometric scales. Xiong et al. [99] developed amphiphilic block copolymer (AB) micelles composed of degradable amphiphilic diblock polymers of poly(ethylene oxide)-block-poly(D,L-lactide)-PEO-b-PDLLA-as nanocarriers for solubilizing 17-AAG and compared its pharmacokinetic behavior with a current formulation of 17-AAG in Cremophor EL (CrEL), ethanol (EtOH), and Polyethylene Glycol (PEG400)-CrEL-EtOH-PEG400. Katragadda et al. [100] developed micellar nanocarriers for the concomitant delivery of paclitaxel and 17-allylamino-17-demethoxygeldanamycin (17-AAG) for cancer therapy by a solvent evaporation method. Our research group has already been studying formulation possibilities using liposomal systems containing 17-AAG on *Leishmania* (L) *amazonensis* amastigotes. We showed promising results in the development of liposomes loaded with 17-AAG:HPβCD, facilitating drug solubilization that can subsequently enhance the distribution of the inhibitor systemically, both orally and intravenously. In fact, this nanoformulation, particularly when incorporated into HPβCD liposomes at 0.006 nM, achieved nearly complete clearance of *L. amazonensis* amastigotes inside macrophages after 48 h of in vitro treatment. These findings underscore the potential of nanotechnology and drug delivery systems to enhance antileishmanial efficacy and reduce the toxicity of 17-AAG. The results support evidence that nanotechnology and drug delivery systems could be used to increase the antileishmanial efficacy and potency of 17-AAG in vitro while also resulting in reduced toxicity that indicates these formulations may represent a potential therapeutic strategy against leishmaniasis [73]. This aids in the administration of the medicine in a real-world context, improving adherence and enabling better monitoring of the treatment of dogs with CVL, and contributes to pharmacovigilance in studies involving the use of medicines in veterinary medicine.

Developing new therapeutic regimens for leishmaniasis treatment is urgent. The rise of drug-resistant strains, high toxicity of existing treatments, co-infections like HIV/Leishmania spp., the limited number of available therapies, and low investment in new drug initiatives are driving researchers and global health agencies to explore innovative strategies for managing and controlling leishmaniasis. These initiatives comprise physical and local therapies (CO_2_ laser administration and thermotherapy, cryotherapy, and electrotherapy), topical drug therapy (nitric oxide derivatives and intralesional drug administration) for cutaneous leishmaniasis and immunomodulators, drug repurposing, and multi-drug or combination therapy for VL [88]. For the past 15 years, trials have explored combination therapies for VL to find shorter, safer regimens that also prevent or delay drug resistance. Combination therapy varies by continent and consists of different therapeutic interventions. Despite promising results from combination therapy trials on the Indian subcontinent, AmBisome monotherapy remains the first-line treatment due to its effectiveness and availability through a donation program, whereas, in Latin America, disappointing trial outcomes have hindered the adoption of combination therapies. Developing an entirely oral combination treatment necessitates the discovery of new chemical entities, many of which are presently being assessed [101,102]. New approaches, such as structure-based drug design (SBDD) and synthetic biology, are promising. SBDD categorizes potential targets by their metabolic pathways and evaluates significant advancements. This approach entails testing compound libraries on either a specific molecular target or an entire organism, typically at the cellular level. Although SBDD studies supplement high-throughput screening and pharmacokinetic optimization, owing to the prevailing view of limited validated targets, certain SBDD strategies have notably contributed to emerging drug candidates with substantial future potential [103]. Synthetic biology involves an interdisciplinary field merging biology, engineering, and computer science, which recently showed potential in developing innovative methods to tackle leishmaniasis [104]. Finally, according to Griensven et al. [102], it is important to emphasize that optimizing the efficacy and extending the lifespan of new treatments necessitates the inclusion of pharmacological substudies for proper dosing and the establishment of systems for the early detection of drug resistance in future research. Regarding 17-AAG, tests combining Hsp90 inhibitors with other potent anti-cancer therapies have shown promising results, not only because of their synergistic antitumor effects but also due to their potential to prolong or prevent the development of drug resistance [86]. These findings open the possibility of testing 17-AAG in combination with other antileishmanial and anti-inflammatory agents for visceral leishmaniasis treatment.

In conclusion, our results demonstrate the potential of 17-AAG in the treatment of CVL under a regimen consisting of three doses at 150 mg/m^2^. This concentration and dose schedule was found to maintain high concentrations of the compound in subcutaneous interstitial fluid with low toxicity levels. Since hepatotoxicity was found to be reversible through this administration protocol, it is possible to propose future efficacy studies using a 150 mg/m^2^ dosage regimen with 48 h intervals. In this context, this study is significant for veterinary medicine and public health as it offers alternative treatments for CVL. Additionally, it facilitates the technological exploration of new formulations containing 17-AAG, enhancing drug administration in real-world scenarios. This improvement promotes better adherence to and monitoring of treatments for dogs with CVL. Moreover, the study contributes to pharmacovigilance by supporting the investigation of drug use in veterinary medicine and other areas.

## 4. Materials and Methods

### 4.1. Preparation of 17-AAG Formulation for Systemic Administration

The systemic delivery formulation of 17-AAG was prepared according to the method described by Burris et al. [22]. Table 2 provides details of the formula used to prepare the 17-AAG solution. The formulation involved two phases: an oily phase and an aqueous phase. Initially, the oily phase was prepared using propylparaben, ethyl alcohol, Cremophor, and 17-AAG. Meanwhile, the aqueous phase was made with methylparaben and propylene glycol. Each phase was mixed separately with agitation for three minutes before the oily phase was incorporated into the aqueous phase under continuous agitation. Subsequently, the pH was measured and adjusted to 7.4 ± 0.1 as needed. The formulation was then filtered through a 0.22 µm syringe filter and stored in a sterile flask at 4 °C until use. At the time of administration, the 17-AAG solution was diluted at a 1:10 ratio in a 0.9% sodium chloride solution. Similarly, a 17-AAG-free placebo formulation was prepared, excluding 17-AAG, and administered to all nine dogs before (time 0) applying the systemic delivery formulation containing 17-AAG (Table 7), as explained in Section 4.3 “Experimental design of pharmacological studies”.

### 4.2. Animal Selection and Care Procedures

Nine dogs, including eight females and one male, without a defined breed, were selected from the kennel of the Gonçalo Moniz Institute (Bahia, Brazil). All eligible animals were submitted to clinical evaluations, and samples (plasma) were collected for hematological and biochemical analysis. Inclusion criteria consisted of age over seven months and clinically healthy status with no history of pharmacological treatment within the previous 15 days. Exclusion criteria were that animals were deemed unsuitable following individual clinical evaluation or laboratory testing, in particular, with altered levels of the liver enzymes, alkaline phosphatase (ALP), alanine aminotransferase (ALT), aspartate aminotransferase (AST), alkaline phosphatase (AF), and gamma-glutamyl transferase (GGT), or aggressivity. 

The kennel is of the open type with natural air renewal, with the bays being screened. The animals were allocated in stalls individually and had a recreation routine at certain times of the day. The animals were fed with super-premium food and free water. All the tasks related to the experiments, which involved the animals, were carried out inside the experimentation kennel to reduce the stress resulting from the animals’ transfer. All experimentation involving canine specimens was performed according to Brazilian federal law for animal experimentation (Law 11794) and the Oswaldo Cruz Foundation’s animal experimentation guidelines (FIOCRUZ). The present study received approval from the Institutional Review Board (CEUA protocol No. 011-2015 and 024-2017) of the Gonçalo Moniz Institute, Bahia-Brazil (IGM–FIOCRUZ/BA).

### 4.3. Experiment Design of Pharmacological Studies

The pharmacological evaluation of 17-AAG was performed after intravenous injections in healthy dogs of 17-AAG with the aid of an infusion pump according to the following studies: Study A: Dose-escalation protocol; and Study B: Multiple administrations at a single dose of 150 mg/m^2^. In both Studies A and B, intravenous administration of the formulations containing 17-AAG was chosen because the majority of studies in both animal models and human clinical trials, as cited in the literature, utilized this administration route [52,55,96,97]. Control animals were the same dogs that, prior to treatment, received an intravenous injection of the placebo solution. After that, samples were collected for hematological and biochemical analyses, as described in Section 4.2, “Animal Selection and Care Procedures”. 

In Study A, 17-AAG was administered in four healthy female dogs to evaluate pharmacokinetics and tolerability. In the first step, before administering a placebo solution free of 17-AAG, as prepared in accordance with Section 4.1, “Preparation of 17-AAG Formulation for Systemic Administration”, each animal underwent clinical assessment. Blood samples were collected for hematological (hemogram) and biochemical analyses, as described in Section 4.2, “Animal Selection and Care Procedures”. This study utilized a 6-step dose-escalation design for the treated dogs. Before 17-AAG administration, the four dogs received a 17-AAG-free placebo solution prepared, as described in Section 4.1, “Preparation of 17-AAG Formulation for Systemic Administration”, and submitted for hematological (hemogram) and biochemical analyses. After a 12-day wash-out period, the dogs received a single dose of 17-AAG at progressively increasing concentrations: 50 mg/m^2^, 100 mg/m^2^, 150 mg/m^2^, 200 mg/m^2^, or 250 mg/m^2^ [81,94]. Wash-out periods, ranging from 7 to 10 days, were observed between each dosage step. Following the systemic administration of 17-AAG, plasma quantification was performed using high-efficiency liquid chromatography (HPLC), as described in Section 4.4.3, “Determination of 17-AAG Concentration by HPLC”. Blood samples were collected at various times, as specified in Section 4.4.1, “Peripheral Blood Sample Collection”. Tolerability evaluations were conducted over 10 days after each 17-AAG administration through daily clinical examinations and blood sampling for laboratory analysis. Peripheral blood samples for hematological and biochemical analyses were collected on days 2, 4, 7, and 10 [105].

Study B involved 9 healthy dogs (8 females and 1 male), including the 4 females from Study A after wash-out and upon confirmation of regular laboratory evaluations. After 12 days, each animal received three doses of 17-AAG at a concentration of 150 mg/m^2^, with intervals of 48 h between doses [105]. For determining the concentration of 17-AAG in the interstitial fluid, samples were collected from three tissue cages that were surgically implanted subcutaneously in the dogs in this study, as described in Section 4.4.2 “Collection of subcutaneous interstitial liquid”. Tolerability evaluations were performed via clinical examinations and blood sampling for laboratory evaluation after each treatment. For toxicity assessments, 4, 6, and 8 h after the final administration, the animals were euthanized for tissue collection and histopathological evaluation. 

### 4.4. 17-AAG Pharmacokinetics Evaluation

#### 4.4.1. Peripheral Blood Sample Collection

In Studies A and B, an intravenous catheter was placed in the cephalic vein of each animal’s right and left anterior limbs. Blood samples were obtained 5 min before the administration of 17-AAG and at several moments after treatment as follows: Study A: 5, 10, 15, 30, 45, 60, and 90 min and 2, 3, 4, 6, 8, 12, 24, and 32 h after administration of 17-AAG; and Study B: 5, 30, and 60 min and 2, 4, 6, 8, 12, 24, and 48 h after 17-AAG administration. Under both protocols, blood samples were centrifuged at 1500× *g* at 4 °C for 10 min. The plasma fraction was stored at −20 °C until the time of 17-AAG quantification in plasma [105].

#### 4.4.2. Collection of Subcutaneous Interstitial Liquid

In Study B, samples of interstitial fluid were collected from tissue cages implanted subcutaneously in the dogs. These cages were prepared based on a protocol previously described by Stegemann et al. [61,62]. They were made from non-toxic, siliconized, flexible, non-irritant PVC tubes with dimensions of 3.5 cm long, an internal diameter of 3 mm, and an external diameter of 5 mm. Three autoclaved cages were surgically implanted between the scapula of each animal under sedation (xylazine 2%, Syntec, Tambore, SP, Brazil) and local anesthesia (lidocaine 2% and epinephrine 0.002%, Syntec). Before and after surgical procedures, antibiotics (enrofloxacin–5 mg/Kg, Syntec) and an anti-inflammatory (ketoprofen 1%–1 mL/Kg, CEVA, Paulínia, SP, Brazil) drug were administered. To allow animals to heal, as well as the formation of the granulation tissue around the cages, a recovery time of 10 days after surgery was considered. The subcutaneous interstitial fluid was collected by percutaneous puncture using a 21 G needle and 20 mL syringe. Samples collected from the tissue cages were centrifuged at 3400× *g* for 10 min and then stored at −20 °C until the time of 17-AAG concentration determination.

#### 4.4.3. Determination of 17-AAG Concentration by HPLC

17-AAG concentrations were determined by HPLC–DAD (Shimadzu, Tokyo, Japan) and a 3 µm, 15 cm × 4.6 mm C18 column (Merck)] in plasma samples obtained under both studies (A and B), as well as in subcutaneous interstitial fluid obtained in Study B. The quantification technique used herein was based on the methods described by Agnew et al. [106] and Johnston et al. [107]. For the mobile phase, acetonitrile (ACN, J.T.Baker^®^, Philadelphia, PA, USA) and water with 1% acetic acid (J.T.Baker^®^)were used. Briefly, aliquots of 200 µL of plasma or 100 µL of subcutaneous interstitial liquid were placed in a 1.5 mL microcentrifuge tube. Subsequently, 600 µL and 300 µL of ACN were added to all plasma and subcutaneous interstitial liquid samples, respectively. The mixture was vortexed for 30 s, followed by centrifugation at 3400× *g* for 10 min. Next, 500 µL of the plasma supernatant and 250 µL of the subcutaneous interstitial fluid were pipetted into a new vial. The method was validated in accordance with the International Council for Harmonization (ICH) technical requirements for the evaluation of medicinal products [108]. To validate the quantification protocol, serial dilutions of 17-AAG (0.012 µg to 10 µg/mL) were added to animal plasma samples, except the negative control. No matrix interferences were observed. Two calibrated plasma curves of 17-AAG were validated with different injection volumes. The first curve was constructed with 17-AAG at concentrations of 125 to 10,000 ng/mL, with an injection volume of 10 µL and the following validation parameters: linearity (R^2^ = 0.9909), intra-day precision (CV ≤ 5%), inter-day precision (CV ≤ 8%), quantification limit of 250 ng/mL, and detection limit of 125 ng/mL. The other curve was constructed with 17-AAG at concentrations of 12 to 300 ng/mL, with an injection volume of 50 µL and the following validation parameters: linearity (R^2^ = 0.9964), intra-day precision (CV ≤ 8%), inter-day precision (CV ≤ 20%), quantification limit of 25 ng/mL, and detection limit of 12.5 ng/mL. Data acquisition and processing were performed using LabSolutions software (Version 5.3).

#### 4.4.4. Determination of 17-AAG Concentration by HPLC

The pharmacokinetic parameters of 17-AAG were obtained from individual concentrations in plasma and subcutaneous interstitial fluid. The AUC was calculated to determine plasma concentrations of 17-AAG from 0 to 8 h after administration (AUC_0–8h_) when the compound’s plasma concentration decreases. The AUC was calculated to determine 17-AAG concentrations in the subcutaneous interstitial fluid during 0–48 h after each administration period (AUC_0–48h_). Cmax, half-life (t ½), distribution volume, and clearance were also determined. Data were analyzed in Excel and WinNonlin using a mono-compartmental model.

### 4.5. 17-AAG Tolerability and Toxicity Assessments

Two separate clinical reports were used to document data obtained from clinical evaluations. A general clinical description was used to note the following information daily: whether the animal was alert, prostrate, depressed, apathetic or aggressive, drinking water, and eating food normally, as well as feces and urine evaluations and the presence of vomit. Every two days, we noted the following parameters: body temperature (°C, measured using a rectal thermometer), heart rate (beats/minute), respiratory rate (cycles/minute), external mucous membrane coloration (conjunctiva and oral), hydration (skin elasticity), superficial lymph node enlargement (submandibular and popliteal), stool consistency, presence of lesions or secretions in the eyes, presence of nasal secretion (quantity and type), coat status, presence of skin lesions and ectoparasites, temperament, presence of claudication, and coordination and balance. Hemograms were performed following blood collection. The following biochemical analyses were performed: total proteins, albumin (A), globulins (Gs), A/G ratio, total cholesterol, bilirubin, renal profile (urea and creatinine), and hepatic enzymatic profiles (ALP, ALT, AST, AF, and GGT). For toxicity assessments, in study B, the animals were euthanized 4, 6, and 8 h after the last administration of 17-AAG, and tissue samples (lung, heart, kidney, liver, spleen, and small and large intestine) were collected for histopathological analysis according to Glaze et al. [82].

### 4.6. Statistical Analysis

Pharmacokinetic parameters of 17-AAG were compared using a paired Student’s *t*-test. Hematological parameters and plasma biochemistry were evaluated with two-way ANOVA, considering time points and doses.

## Figures and Tables

**Figure 1 pharmaceuticals-17-00767-f001:**
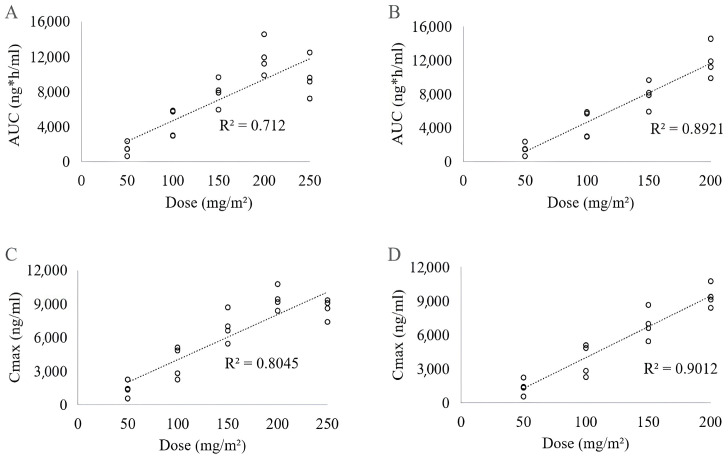
Relationship between doses of 17-AAG and plasma AUC or Cmax following intravenous administration in dogs. (**A**,**C**) Doses from 50 to 250 mg/m^2^; (**B**,**D**) Doses from 50 to 200 mg/m^2^.

**Figure 2 pharmaceuticals-17-00767-f002:**
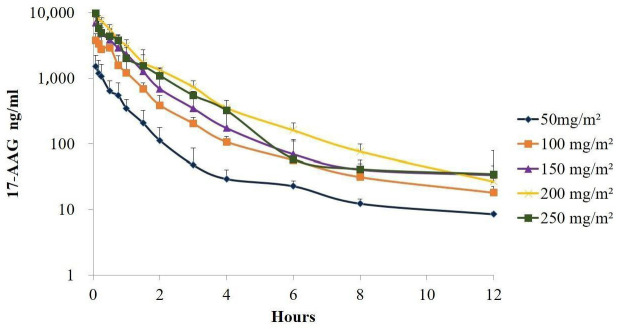
Plasma concentration levels of various doses of 17-AAG were assessed for up to 12 h following intravenous administration through a 30 min continuous infusion in dogs.

**Figure 3 pharmaceuticals-17-00767-f003:**
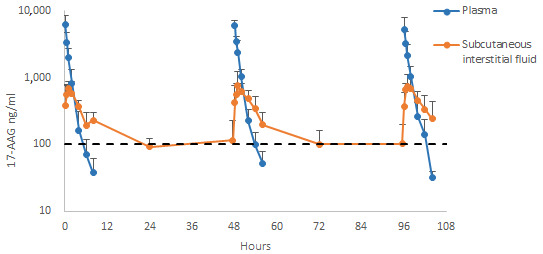
Concentrations of 17-AAG in plasma and subcutaneous interstitial fluid in intravenously treated dogs. This figure shows the levels of 17-AAG in plasma and subcutaneous interstitial fluid of dogs, measured up to 104 h after administering multiple intravenous doses of 150 mg/m^2^. The dashed lines represent the 17-AAG concentration previously proven effective against Leishmania promastigotes in in vitro studies.

**Figure 4 pharmaceuticals-17-00767-f004:**
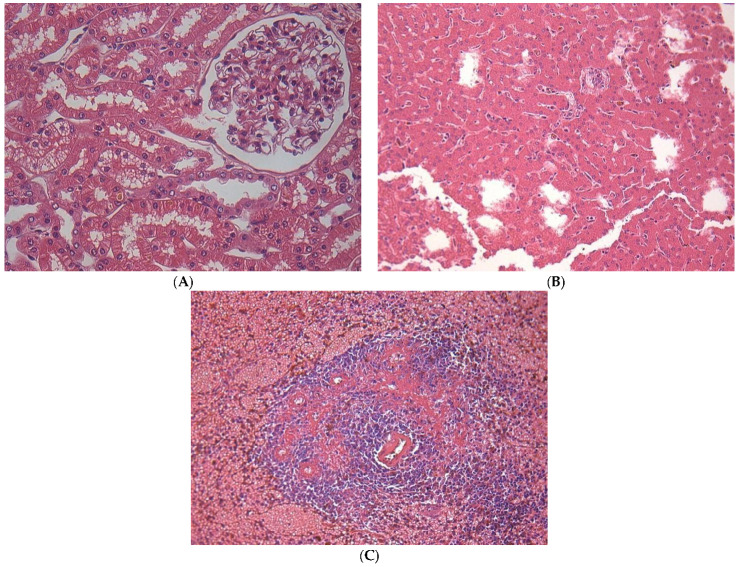
Histological images depicting alterations in various organs of dogs treated with 17-AAG. Animals were treated with three intravenous administrations of 150 mg/m2 of 17-AAG at 48 h intervals. (**A**) Shows vacuolar degeneration of the renal tubular epithelium. (**B**) Illustrates mobilization of Kupffer cells in the liver. (**C**) Displays plasmocytic hyperplasia and disorganization of type II/III in the spleen. All images are at 40× magnification.

**Figure 5 pharmaceuticals-17-00767-f005:**
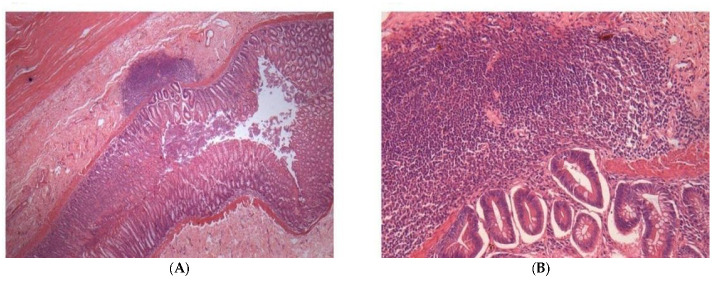
Histological images depicting alterations in the large intestine of dogs treated with 17-AAG, as described in Figure 2. Panels (**A**,**B**) show the presence of colitis in the treated dogs. Image (**A**) is at 4× magnification, and image (**B**) is at 40× magnification.

**Table 1 pharmaceuticals-17-00767-t001:** Pharmacokinetic parameters of 17-AAG in dogs according to different intravenously administered doses.

Dose (mg/m^2^)	AUC (ng × h/mL)	Cmax (ng/mL)	t½ (h)	Clearance (L/m^2^/h)	DV (L/m^2^)
	Mean ± SD	Mean ± SD	Mean ± SD	Mean ± SD	Mean ± SD
50	1483.26 ± 694.52	1405.97 ± 686.71	0.54 ± 0.03	0.04 ± 0.02	0.03 ± 0.02
100	4380.35 ± 1626.22	3756.41 ± 1422.52	0.61 ± 0.08	0.03 ± 0.01	0.02 ± 0.01
150	7927.84 ± 1548.51	6938.99 ± 1342.39	0.59 ± 0.04	0.02 ± 0.00	0.02 ± 0.00
200	11,902.75 ± 1962.12	9439.70 ± 991.11	0.67 ± 0.05	0.02 ± 0.00	0.02 ± 0.00
250	9632.29 ± 2667.12	8611.51 ± 1062.88	0.57 ± 0.21	0.03 ± 0.01	0.02 ± 0.00

AUC = area under the curve as a function of time; Cmax = maximum concentration; t½ = half-life time; DV = distribution volume; SD = standard deviation.

**Table 2 pharmaceuticals-17-00767-t002:** Side and adverse (alterations) effects observed in healthy dogs following the intravenous administration of different concentrations of 17-AAG.

Alterations	Number of Animals Evaluated (*n* = 4) N (%)				
	50 mg/m^2^	100 mg/m^2^	150 mg/m^2^	200 mg/m^2^	250 mg/m^2^
**Biochemical**					
ALT	1 (25%)	2 (50%)	3 (75%)	3 (75%)	4 (100%)
AST	0	2 (50%)	2 (50%)	4 (100%)	4 (100%)
**Gastrointestinal**					
Bloody diarrhea	0	0	0	3 (75%)	4 (100%)
Diarrhea (pasty feces)	0	1 (25%)	1 (25%)	1 (25%)	1 (25%)
Nausea	0	0	0	4 (100%)	4 (100%)
Vomit	0	0	0	1 (25%)	1 (25%)
**Other**					
Hyperthermia	0	0	1 (25%)	0	1 (25%)
Pruritus	0	0	1 (25%)	3 (75%)	4 (100%)
Erythema	0	0	1 (25%)	2 (50%)	4 (100%)

**Table 3 pharmaceuticals-17-00767-t003:** Pharmacokinetic parameters of 17-AAG in plasma of dogs treated with 17-AAG *.

Administration	t ½ (h)	Cmax (μg/mL)	Clearance (L/m^2^/h)	DV (L/m^2^)	AUC (μg/mL × h) 0–8 h
	Mean ± SD	Mean ± SD	Mean ± SD	Mean ± SD	Mean ± SD
1st	0.69 ± 0.22	6309 ± 2204	0.023 ± 0.002	0.029 ± 0.003	6353 ± 432
2nd	0.69 ± 0.24	5912 ± 1127	0.021 ± 0.001	0.030 ± 0.002	7054 ± 382
3rd	0.81 ± 0.35	5254 ± 2784	0.022 ± 0.002	0.033 ± 0.003	6850 ± 69

* Dogs received three intravenous administrations of 150 mg/m^2^ of 17-AAG at 48 h intervals. AUC = area under the curve as a function of time; Cmax = maximum concentration; t½ = half-life time; DV = distribution volume; SD = standard deviation.

**Table 4 pharmaceuticals-17-00767-t004:** Pharmacokinetic parameters of 17-AAG in the subcutaneous interstitial fluid after treatment *.

Administration	t ½ (h)	Cmax (μg/mL)	AUC (μg/mL × h)				
			0–4 h	0–6 h	0–8 h	0–24 h	0–47.5 h
	Mean ± SD	Mean ± SD	Mean ± SD	Mean ± SD	Mean ± SD	Mean ± SD	Mean ± SD
1st	38 ± 16	704 ± 332	1820 ± 430	2301 ± 409	2659 ± 437	4765 ± 333	7382 ± 1357
2nd	28 ± 9	736 ± 294	1943 ± 618	2553 ± 638	2913 ± 686	4070 ± 700	7054 ± 2189
3rd	N/A	726 ± 401	895 ± 408	2040 ± 1097	2734 ± 1442	3505 ± 2417	N/A

* Dogs received three intravenous administrations of 150 mg/m^2^ of 17-AAG at 48 h intervals. AUC = area under the curve as a function of time; Cmax = maximum concentration; t½ = half-life time; SD = standard deviation; N/A: unvalued.

**Table 5 pharmaceuticals-17-00767-t005:** Side and adverse (alterations) effects in dogs after multiple administrations of 17-AAG *.

Alterations	Animals Presenting Adverse Reactions *n* (%)
**Biochemistry**	
ALT	3 (33%)
AST	3 (33%)
**Gastrointestinal**	
Diarrhea (liquid feces)	3 (33%)
Pasty feces	9 (100%)
Bloody diarrhea	1 (11%)
Nausea	0
Vomit	0
**Others**	
Hyperthermia	0
Skin reaction	1 (11%)
Erythema	0
Pruritus	0

* Dogs received three intravenous administration of 17-AAG at 150 mg/m^2^ with 48 h intervals.

**Table 6 pharmaceuticals-17-00767-t006:** Histopathological changes in tissues from dogs after multiple administrations of 17-AAG.

Histopathological Alterations	Animals Presenting Alterations *n* (%)
**Kidney**	
Vacuolar degeneration of the renal tubular epithelium	9 (100%)
**Liver**	
Mobilization of Kupffer cells	9 (100%)
**Spleen**	
Congestion	9 (100%)
Immunoblastic hyperplasia	9 (100%)
Spleen disorganization in type II/III	9 (100%)
**Small Intestine**	
Mucosal calcification foci	2 (22%)
**Thick Intestine**	
Mucosal calcification foci	1 (11%) 5 (55%)
Colitis	

**Table 7 pharmaceuticals-17-00767-t007:** Ingredients used in the formulation of 17-AAG for systemic administration.

Reagent	Quantity
17-AAG or Tanespimycin (LC-Laboratories^®^, Woburn, MA, USA)	0.1 g
Cremophor (BASF^®^, Cincinnati, OH, USA)	2.1 g
Propylene glycol (Palmar^®^, Salvador, BA, Brazil)	3.0 g
Methylparaben (Dinamica^®^, Recreio Campestre Jóia, Indaiatuba, SP, Brazil)	0.002 g
Propylparaben (Dinamica^®^)	0.0002 g
Ethyl alcohol (Merck^®^, Darmstadt, Germany)	4.9 mL

## Data Availability

Data are contained within the article.

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
