# Peer review of "Pharmacokinetics, Dose-Proportionality, and Tolerability of Intravenous Tanespimycin (17-AAG) in Single and Multiple Doses in Dogs: A Potential Novel Treatment for Canine Visceral Leishmaniasis"

_pharmaceuticals, 2024, doi:10.3390/ph17060767_

Round 1

Reviewer 1 Report

Comments and Suggestions for Authors

"Pharmacokinetics, Dose-Proportionality, and Tolerability of Intravenous Tanespimycin (17-AAG) in Single and Multiple Doses in Dogs: A Potential Novel Treatment for Canine Visceral Leishmaniasis" is a comprehensive study that evaluates the safety, pharmacokinetics, and dose-proportionality of 17-AAG in dogs, to explore its potential as a treatment for canine visceral leishmaniasis (CVL). The study addresses a significant gap in the treatment of CVL, a disease with high incidence rates in Brazil and other countries. The investigation of 17-AAG as a potential treatment is innovative and could have substantial impacts on public and veterinary health.

 This review report offers suggestions for improvement for the authors.

While the study is well-designed, the manuscript could benefit from a more precise presentation of comparative data between treated and control groups, particularly regarding tolerability and safety assessments. This would strengthen the conclusions drawn regarding 17-AAG's safety and efficacy.

 Further investigation into the ramifications of these findings, possible harmful processes, and their relevance to clinical application could be conducted considering the hepatotoxic effects reported at greater doses.

- Consideration of Alternative Administration Routes: The manuscript focuses on intravenous administration. Including discussions on the feasibility, advantages, or limitations of alternative administration routes as future perspectives (oral, subcutaneous) for 17-AAG could enhance the paper's comprehensiveness.

  1. Provide a more precise explanation for the dose-escalation study's chosen dose range (50–250 mg/m²) and the rationale behind selecting 150 mg/m² for multiple dose administrations. Discuss any previous studies or preliminary data that informed these choices.
  2.       Provide more details about the control group, including what they were given and how they were treated, focusing on using placebos. Describe the steps to ensure the outcomes were evaluated without bias or blinding.

Add a more in-depth assessment of the treatment and control groups' impacts, contrasting safety and efficacy outcomes through statistical analyses.

Provide a more in-depth discussion on hepatotoxicity, including potential mechanisms, comparison with other treatments for CVL, and how these effects might be mitigated in a clinical setting.

- Discuss potential alternative administration routes for 17-AAG, considering the study's findings and how they might influence drug formulation and delivery in a real-world context.

Improve the clarity of data presentation, particularly in tables and figures, to enhance readability and comprehension of the pharmacokinetic parameters and histopathological findings.

Suggest future research directions, including studies on long-term effects, combination therapies with other antileishmanial drugs, and exploration of dose optimization to minimize toxicity while maintaining efficacy.

- Ensure that, When used for the first time, technical terms and acronyms are defined clearly. This will enhance the manuscript's accessibility to readers unfamiliar with the field.

  • Pay attention to the use of definite (the) and indefinite (a, an) articles, especially when introducing new concepts or specific items (e.g., "the Hsp90 inhibitor" vs. "an Hsp90 inhibitor").

Comments on the Quality of English Language

The manuscript is well-written but could benefit from targeted improvements in clarity, grammar, coherence, and technical accuracy to ensure it meets the high standards of international scientific communication.

Author Response

Reviewer #1 

"Pharmacokinetics, Dose-Proportionality, and Tolerability of Intravenous Tanespimycin (17-AAG) in Single and Multiple Doses in Dogs: A Potential Novel Treatment for Canine Visceral Leishmaniasis" is a comprehensive study that evaluates the safety, pharmacokinetics, and dose-proportionality of 17-AAG in dogs, to explore its potential as a treatment for canine visceral leishmaniasis (CVL). The study addresses a significant gap in the treatment of CVL, a disease with high incidence rates in Brazil and other countries. The investigation of 17-AAG as a potential treatment is innovative and could have substantial impacts on public and veterinary health.

While the study is well-designed, the manuscript could benefit from a more precise presentation of comparative data between treated and control groups, particularly regarding tolerability and safety assessments. This would strengthen the conclusions drawn regarding 17-AAG's safety and efficacy.

  1. Further investigation into the ramifications of these findings, possible harmful processes, and their relevance to clinical application could be conducted considering the hepatotoxic effects reported at greater doses.

 This review report offers suggestions for improvement for the authors.

Answer - We appreciate the reviewer's positive feedback on our manuscript. Additionally, we believe their insightful comments will further enhance the paper, making it more suitable for publication in the special issue of Pharmaceuticals titled 'Recent Advancements in the Development of Antiprotozoal Agents'

  1. Consideration of Alternative Administration Routes: The manuscript focuses on intravenous administration. Including discussions on the feasibility, advantages, or limitations of alternative administration routes as future perspectives (oral, subcutaneous) for 17-AAG could enhance the paper's comprehensiveness.

Answer - We appreciate the reviewer's valuable suggestion. To address this point in the text, we have included the following statement in the Discussion section to explore the possibility of an alternative route for administering 17-AAG to treat dogs naturally infected with Leishmania infantum.

OK Line 611-654: The major obstacle for delivery of 17-AAG related to its limited aqueous solubility (ca. 0.01 mg/mL) can encourage studies of special formulations for better solubilization and delivery of the drug, by various administration routes, including oral. Although oral administration seems to be a viable route for treatment, given that leishmaniasis is a systemic disease affecting multiple organs, the majority of literature involving 17-AAG, in both animal models and human studies, predominantly uses intravenous administration [54,9497]. According to the National Institute of Cancer's guide, "Turning Molecules into Medicines for Public Health https://dtp.cancer.gov/timeline/posters/AGG_Geldamycin.pdf," mice with MCF-7 breast tumors treated with various doses of 17-AAG exhibited a maximum plasma half-life of 4.4 hours following a 40 mg/kg intravenous dose, yet the drug was undetectable after oral administration. Biopharmaceutical strategies can be useful in this process, such as the incorporation of 17-AAG into micelles with special polymers, liposomes and other delivery systems, on nanometric scales. Xiong et al. [98] developed amphiphilic block copolymer (AB) micelles composed of degradable amphiphilic diblock polymers of poly(ethylene oxide)-block-poly(D,L-lactide) - PEO-b-PDLLA - as nanocarriers for solubilizing 17-AAG, and compared its pharmacokinetic behavior with a current formulation of 17-AAG in Cremophor EL (CrEL), ethanol (EtOH) and Polyethylene Glycol (PEG400) - CrEL-EtOH-PEG400. Katragadda et al. [99] developed micellar nanocarriers for concomitant delivery of paclitaxel and 17-allylamino-17-demethoxygeldanamycin (17-AAG) for cancer therapy by a solvent evaporation method. Our research group has already been studying formulation possibilities, using liposomal systems containing 17-AAG on Leishmania (L) amazonensis amastigotes. We showed promising results in the development of liposomes loaded with 17-AAG:HPβCD, facilitating drug solubilization that can subsequently enhance the distribution of the inhibitor systemically, both orally and intravenously. In fact, this nanoformulation, particularly when incorporated into HPβCD liposomes at 0.006 nM, achieved nearly complete clearance of L. amazonensis amastigotes inside macrophages after 48 hours of in vitro treatment. These findings underscore the potential of nanotechnology and drug delivery systems to enhance the antileishmanial efficacy and reduce the toxicity of 17-AAG. The results support evidence that nanotechnology and drug delivery systems could be used to increase the antileishmanial efficacy and potency of 17-AAG in vitro, while also resulting in reduced toxicity that indicates these formulations may represent a potential therapeutic strategy against leishmaniasis [100]. This aids in the administration of the medicine in a real-world context, improving adherence and enabling better monitoring of the treatment of dogs with CVL, and contributes to pharmacovigilance in studies involving the use of medicines in veterinary medicine.

In the Materials and Methods section, we have also included an explanation for choosing intravenous administration of the 17-AAG formulation as follows:

OK Lines 774-781: In both Studies A and B, intravenous administration of the formulations containing 17-AAG was chosen because the majority of studies in both animal models and human clinical trials, as cited in the literature, have utilized this administration route [51,54,95,96]. Control animals were the same dogs that, prior to treatment, received an intravenous injection of the placebo solution. After that, samples were collected for hematological and biochemical analyses described in Section 4.2 Animal selection and care procedures

  1. Provide a more precise explanation for the dose-escalation study's chosen dose range (50–250 mg/m²) and the rationale behind selecting 150 mg/m² for multiple dose administrations. Discuss any previous studies or preliminary data that informed these choices. 

Answer - The study's dose-escalation range (50–250 mg/m²) was chosen based on studies in humans evaluating different dosing schedules of single agent 17-AAG in adults and children (Nowakowski et al., 2006*). In this study, the dose escalation tested was 57-308 mg/m². Given that the maximum tolerated dose in this study was 220 mg/m² per dose, we opted for the dose-escalation range of 50–250 mg/m².

*Nowakowski GS, McCollum AK, Ames MM et al (2006) A phase I trial of twice-weekly 17-allylamino-demethoxy-geldanamycin in patients with advanced cancer. Clin Cancer Res 12:6087–6093.

The decision to administer 150 mg/m² for multiple dose administrations stemmed from the findings of Study A. Within this study, the administration of 17-AAG at concentrations ranging from 50 to 150 mg/m² demonstrated a favorable safety profile, with minimal occurrence of side effects. Following the establishment of a maximum tolerable dose of 150 mg/m² of 17-AAG in canines during Study A, a subsequent investigation (Study B) was undertaken, employing three doses of 150 mg/m² with 48-hour intervals between administrations.

  1. Provide more details about the control group, including what they were given and how they were treated, focusing on using placebos. Describe the steps to ensure the outcomes were evaluated without bias or blinding.

Answer - Referee highlighted a critical issue concerning the need for a clearer explanation of the control group. In response, we have added detailed explanations about the placebo formulation and  control animals throughout the Materials and Methods section.

OK Lines 730-734: 4.1 Preparation of 17-AAG formulation for systemic administration

Similarly, a 17-AAG-free placebo formulation was prepared, excluding 17-AAG, and administered to all nine dogs before (time 0) applying the systemic delivery formulation containing 17-AAG as explained in the 4.3 Experimental design of pharmacological studies.

OK Lines 780-811: Control animals were the same dogs that, prior to treatment, received an intravenous injection of the placebo solution. After that, samples were collected for hematological and biochemical analyses described in Section 4.2 Animal selection and care procedures. 

In Study A, 17-AAG was administered in 4 healthy female dogs to evaluate pharmacokinetics and tolerability. In the first step, before administering a placebo solution free of 17-AAG, as prepared in accordance with Section 4.1, 'Preparation of 17-AAG Formulation for Systemic Administration,' each animal underwent clinical assessment. Blood samples were collected for hematological (hemogram) and biochemical analyses described in Section 4.2 Animal selection and care procedures. This study utilized a 6-step dose-escalation design for the treated dogs, Before 17-AAG administration, the four dogs received a 17-AAG-free placebo solution, prepared as described in Section 4.1, "Preparation of 17-AAG Formulation for Systemic Administration" and also submitted for hematological (hemogram) and biochemical analyses. After a 12-day wash-out period, the dogs received a single dose of 17-AAG at progressively increasing concentrations: 50 mg/m², 100 mg/m², 150 mg/m², 200 mg/m², or 250 mg/m² [80,93]. Wash-out periods, ranging from 7 to 10 days, were observed between each dosage step. Following systemic administration of 17-AAG, plasma quantification was performed using high-efficiency liquid chromatography (HPLC), as described in Section 4.4.3, "Determination of 17-AAG Concentration by HPLC." Blood samples were collected at various times as specified in Section 4.4.1, "Peripheral Blood Sample Collection." Tolerability evaluations were conducted over 10 days after each 17-AAG administration through daily clinical examinations and blood sampling for laboratory analysis. Peripheral blood samples for hematological and biochemical analyses were collected on days 2, 4, 7, and 10 [104].

Study B involved 9 healthy dogs (8 females and 1 male), including the 4 females from Study A after wash-out and upon confirmation of regular laboratory evaluations. 

  1. Add a more in-depth assessment of the treatment and control groups' impacts, contrasting safety and efficacy outcomes through statistical analyses.

Answer: We acknowledge the referee's concerns regarding the statistical analyses of the treatment impact on several parameters evaluated in this study, which are addressed throughout the text. Additionally, a "Statistical Analysis" subsection has been added to the "Materials and Methods" section as follows:

OK Line 150-166: Significant differences in AUC and Cmax were observed when comparing doses of 50, 100, 150, and 200 mg/m², as well as between 50, 100, and 250 mg/m². However, no differences were found between doses of 150 or 200 mg/m² and 250 mg/m². We only found differences between half-life (t½) values between doses of 150 and 200 mg/m². Clearance of 17-AAG from plasma was, on average, 0.026 ± 0.010 L/h/m², with a coefficient of variation (CV) of 36%, while the CV between animals ranged from 1% to 20% that were not statistically different among doses. Similarly, no significant differences were observed among the various doses for distribution volume (DV) (Table 1). Figures 1 and Table 1 respectively illustrate that both AUC (measured over time) and Cmax increased proportionally with administered doses of 17-AAG, showing a linear correlation between doses ranging from 50 to 200 mg/m2. However, no correlation was observed between plasma concentration and a dose of 250 mg/m² (Figure 2). Also, we observed a rapid decline of 17-AAG in plasma regardless of the dose applied (Figure 2). 

OK Lines 280-288: Regarding the pharmacokinetic differences among dogs given three doses of 150 mg/m2 of 17-AAG, a significant difference was observed only in the t½ between the first and third administrations. No changes in Cmax values were observed in either plasma or interstitial fluid at any of the assessments undertaken after each 150 mg/m2 dose. Significant differences in AUC were observed when comparing the time intervals of 0-8 hours with 0-24 hours, and 0-6 hours with 0-24 hours. No significant differences were found among other intervals or across different administrations (Tables 3 and 4).

OK Lines 320-338: When considering hematological parameters and plasma biochemistry in animals after the administration of 17-AAG , only the total protein showed a significant difference among dogs in relation to drug concentration. Dogs receiving higher doses exhibited elevated levels of total protein in their serum. Plasma biochemistry analyses revealed that the enzymes, alanine aminotransferase (ALT) (28.57 ± 4.29 to 173.33 ± 49.56 U/L), aspartate aminotransferase (AST) (27.85 ± 3.80 to 248.20 ± 85.80 U/L), and Gamma-Glutamyl Transferase (GGT) (1.60 ± 0.06 to 12.70 ± 0.50 U/L) showed to be elevated in treated dogs but these differences revealed not to be statically distinct and reversible after drug washout.

When analyzing the frequency of side effects in dogs receiving varying doses of 17-AAG, increases in AST were noted in the plasma of dogs treated with 100 mg/m² (50% of animals), 150 mg/m² (50%), 200 mg/m² (100%), and 250 mg/m² (100%). However, these elevations were reversible. In animals where multiple doses of 17-AAG at 150 mg/m² were administered, elevated levels of ALT and AST were observed in only 33% of the dogs (3 out of 9). By the fourth day, elevated transaminase levels persisted in only one animal (Table 5).

OK Lines: 925-929 in Material and methods Section:

4.6. Statistical analysis

Pharmacokinetic parameters of 17-AAG were compared using a paired Student’s t-test. Hematological parameters and plasma biochemistry were evaluated with a two-way ANOVA, considering time points and doses.

  1. Provide a more in-depth discussion on hepatotoxicity, including potential mechanisms,  comparison with other treatments for CVL, and how these effects might be mitigated in a clinical setting.

Answer - In response to the referee's comment, in fact hepatotoxicity represents a significant limitation of 17-AAG treatment that warrants further investigation. Accordingly, we have included a paragraph summarizing this information in the Discussion section

OK Line 538-592: Models involving rats and dogs treated with 17-AAG indicated that the liver is a target organ for mild- to moderate lesions, leading to increases in AST, ALT, alkaline phosphatase and GGT in dogs [75,82]. Although it is a less toxic derivative of geldanamycin, 17-AAG still causes serious toxic effects, including hepatotoxicity. In a phase I study conducted by Ramanathan et al. [54], dose-limiting toxicities including hepatotoxicity were observed, along with side effects such as vomiting, nausea, anemia, and myalgias. In addition, histopathological analysis indicated Kupffer cell mobilization in the livers of all 9 animals treated with 3 doses of 17-AAG at 150 mg/m². We suggest that these alterations likely occurred due to an enhancement in liver metabolism secondary to increased cytochrome P450 activity [51,83]. Amin et al. [84] evaluated the toxicity of 17-AAG in canine liver fragments incubated in a medium containing 17-AAG at concentrations ranging from 0.1 to 5 μM. They found that inhibition of epithelial cell proliferation occurs in a concentration and time-dependent manner and altered levels of AST, ALT, ALP and GGT in treated fragments. The mechanism related to liver toxicity caused by geldanamycin and its derivatives appears to depend on the benzoquinone moiety of these compounds [85]. However, the precise mechanisms underlying 17-AAG's hepatotoxicity require further investigation. Samuni et al. [86] suggested that the toxicity against rat primary hepatocytes induced by these Hsp90 inhibitors could be attributed to the generation of reactive oxygen species, such as superoxide.

Among drugs commonly used as first-line treatments for VL, hepatotoxicity has been reported for pentavalent antimonials and the liposomal formulation of amphotericin B (L-AMB). Like 17-AAG, antimonials cause hepatotoxicity along with other severe side effects, including cardiotoxicity and pancreatitis, and symptoms like vomiting, nausea, anorexia, myalgia, and abdominal pain [87]. Kato et al. [88] described that the mechanism associated with this side effect might involve the accumulation of residual Sb(III) from meglumine antimoniate and that co-treatment with ascorbic acid could reduce hepatic alterations caused by antimonials in infected mice. Their findings support strategies to reduce liver toxicity associated with antimonial therapy in humans using pentavalent antimonials with minimal Sb(III) residue supplemented with ascorbic acid. Hepatotoxicity from treatment with L-AMB was observed in 21% of patients in a retrospective study, potentially related to the compound's high affinity for binding to biological membranes and lipoproteins, leading to the accumulation of AMB in the liver. This accumulation may elevate transaminase or bilirubin levels, potentially causing organ failure [89].  In contrast to the other drugs in use, the oral treatment of leishmaniasis with miltefosine, which is generally well-tolerated in a monotherapy regimen, primarily causes mild gastrointestinal issues, and no reported occurrence of hepatotoxicity in either animal models or humans. Similar to the effects seen with 17-AAG in dogs, miltefosine can cause mild-to-moderate elevations in transaminase levels, which typically return to normal upon dosage reduction [90]. Like Glaze et al. [81], the present report found congestion and atrophy in spleen lymphoid tissue and inflammatory lesions in the small and large intestines, in all 9 dogs that received three doses of 150 mg/m² during Study B protocol. While renal toxicity had only been previously described in mice [82], this is the first report that described a vacuolar degeneration in the tubular epithelium in the kidneys of all 9 dogs from Study B.

  1. Discuss potential alternative administration routes for 17-AAG, considering the study's findings and how they might influence drug formulation and delivery in a real-world context. 

Answer - We appreciate the reviewer's valuable suggestion. While 17-AAG is less toxic than Geldanamycin, its use still poses several challenges. Formulating 17-AAG is difficult, and patient administration is often problematic due to its garlic-like odor. Additionally, when 17-AAG is in a free formulation, it is predominantly administered intravenously, which is not ideal due to its suboptimal pharmaceutical properties. To address these issues and present alternatives, we have added the following paragraph to the Discussion section.

OK Line 611-654: The major obstacle for delivery of 17-AAG related to its limited aqueous solubility (ca. 0.01 mg/mL) can encourage studies of special formulations for better solubilization and delivery of the drug, by various administration routes, including oral. Although oral administration seems to be a viable route for treatment, given that leishmaniasis is a systemic disease affecting multiple organs, the majority of literature involving 17-AAG, in both animal models and human studies, predominantly uses intravenous administration [54,9497]. According to the National Institute of Cancer's guide, "Turning Molecules into Medicines for Public Health https://dtp.cancer.gov/timeline/posters/AGG_Geldamycin.pdf," mice with MCF-7 breast tumors treated with various doses of 17-AAG exhibited a maximum plasma half-life of 4.4 hours following a 40 mg/kg intravenous dose, yet the drug was undetectable after oral administration. Biopharmaceutical strategies can be useful in this process, such as the incorporation of 17-AAG into micelles with special polymers, liposomes and other delivery systems, on nanometric scales. Xiong et al. [98] developed amphiphilic block copolymer (AB) micelles composed of degradable amphiphilic diblock polymers of poly(ethylene oxide)-block-poly(D,L-lactide) - PEO-b-PDLLA - as nanocarriers for solubilizing 17-AAG, and compared its pharmacokinetic behavior with a current formulation of 17-AAG in Cremophor EL (CrEL), ethanol (EtOH) and Polyethylene Glycol (PEG400) - CrEL-EtOH-PEG400. Katragadda et al. [99] developed micellar nanocarriers for concomitant delivery of paclitaxel and 17-allylamino-17-demethoxygeldanamycin (17-AAG) for cancer therapy by a solvent evaporation method. Our research group has already been studying formulation possibilities, using liposomal systems containing 17-AAG on Leishmania (L) amazonensis amastigotes. We showed promising results in the development of liposomes loaded with 17-AAG:HPβCD, facilitating drug solubilization that can subsequently enhance the distribution of the inhibitor systemically, both orally and intravenously. In fact, this nanoformulation, particularly when incorporated into HPβCD liposomes at 0.006 nM, achieved nearly complete clearance of L. amazonensis amastigotes inside macrophages after 48 hours of in vitro treatment. These findings underscore the potential of nanotechnology and drug delivery systems to enhance the antileishmanial efficacy and reduce the toxicity of 17-AAG. The results support evidence that nanotechnology and drug delivery systems could be used to increase the antileishmanial efficacy and potency of 17-AAG in vitro, while also resulting in reduced toxicity that indicates these formulations may represent a potential therapeutic strategy against leishmaniasis [100]. This aids in the administration of the medicine in a real-world context, improving adherence and enabling better monitoring of the treatment of dogs with CVL, and contributes to pharmacovigilance in studies involving the use of medicines in veterinary medicine.

Also the following sentence was added to conclusion:

OK Lines 706-713: In this context, this study is significant for Veterinary Medicine and Public Health as it offers alternative treatments for CVL. Additionally, it facilitates the technological exploration of new formulations containing 17-AAG, enhancing drug administration in real-world scenarios. This improvement promotes better adherence to and monitoring of treatments for dogs with CVL. Moreover, the study contributes to Pharmacovigilance by supporting the investigation of drug use in Veterinary Medicine and other areas.

  1.  Improve the clarity of data presentation, particularly in tables and figures, to enhance readability and comprehension of the pharmacokinetic parameters and histopathological findings.

Answer - This is an important point raised by the reviewer. Please refer throughout the text to the tables and graphs we modified format and legend content according to the reviewer's suggestions. Please note that the figures have been renumbered and redistributed according to their appearance in the text.

  1. Suggest future research directions, including studies on long-term effects, combination therapies with other antileishmanial drugs, and exploration of dose optimization to minimize toxicity while maintaining efficacy.

Answer - We appreciate the reviewer's valuable suggestion. A paragraph has been included in the Discussion section as follows.

OK Line 656-698: Developing new therapeutic regimens for leishmaniasis treatment is urgent. The rise of drug-resistant strains, high toxicity of existing treatments, co-infections like HIV/Leishmania spp., a limited number of available therapies, and low investment in new drug initiatives are driving researchers and global health agencies to explore innovative strategies for managing and controlling leishmaniasis. These initiatives comprise physical and local therapies (CO2 laser administration and thermotherapy, cryotherapy, electrotherapy), topical drug therapy (nitric oxide derivatives, intralesional drug administration) for cutaneous leishmaniasis, and immunomodulators, drug repurposing, and multi-drug or combination therapy for VL [87]. For the past 15 years, trials have explored combination therapies for VL to find shorter, safer regimens that also prevent or delay drug resistance. Combination therapy varies by continent and consists of different therapeutic interventions. Despite promising results from combination therapy trials on the Indian subcontinent, AmBisome monotherapy remains the first-line treatment due to its effectiveness and availability through a donation program, whereas in Latin America, disappointing trial outcomes have hindered the adoption of combination therapies. Developing an entirely oral combination treatment necessitates the discovery of new chemical entities, many of which are presently being assessed [101]. New approaches, such as structure-based drug design (SBDD) and synthetic biology, are promising. SBDD categorizes potential targets by their metabolic pathways and evaluates significant advancements. This approach entails testing compound libraries on either a specific molecular target or an entire organism, typically at the cellular level. Although SBDD studies supplement high-throughput screening and pharmacokinetic optimization, owing to the prevailing view of limited validated targets, certain SBDD strategies have notably contributed to emerging drug candidates with substantial future potential [102]. Synthetic biology involves an interdisciplinary field merging biology, engineering, and computer science, which, recently, showed potential in developing innovative methods to tackle leishmaniasis [103]. Finally, according to Griensven et al. [101], it is important to emphasize that optimizing the efficacy and extending the lifespan of new treatments necessitates the inclusion of pharmacological substudies for proper dosing and the establishment of systems for early detection of drug resistance in future research. Regarding 17-AAG, tests combining Hsp90 inhibitors with other potent anticancer therapies have shown promising results, not only because of their synergistic antitumor effects but also due to their potential to prolong or prevent the development of drug resistance [85]. These findings open the possibility of testing 17-AAG in combination with other antileishmanial and anti-inflammatory agents for visceral leishmaniasis treatment.

  1. Ensure that, When used for the first time, technical terms and acronyms are defined clearly. This will enhance the manuscript's accessibility to readers unfamiliar with the field. 

Answer - We sincerely appreciate the reviewer’s comments. In response to the feedback, we have provided detailed definitions for the acronyms (AUC, ALT, AST, GGT, PK, PK/PD, and DMSO) in the text.

  1. Pay attention to the use of definite (the) and indefinite (a, an) articles, especially when introducing new concepts or specific items (e.g., "the Hsp90 inhibitor" vs. "an Hsp90 inhibitor"). The manuscript is well-written but could benefit from targeted improvements in clarity, grammar, coherence, and technical accuracy to ensure it meets the high standards of international scientific communication.

Answer - We sincerely appreciate the reviewer’s feedback. Please refer to alterations have been embedded throughout the text.

Reviewer 2 Report

Comments and Suggestions for Authors

Marcos Ferrante et al. presented pharmacokinetics (PK), dose linearity, and tolerability of intravenous Tanespimycin (17-AAG) in single and multiple dogs. The author demonstrated the potential of 17-AAG in the treatment of CVL, using a regimen of three doses at 150 mg/m². Please, find below my comments.

1.     The author can show the PK/PD graph to get a more quantitative sense of the data interpretation.

2.     As this is a high protein-bound compound, please use protein-adjusted IC50 in showing the PK/PD relationship.

3.     Throughout the manuscript, I didn’t see any statistical comparison of the results. Please, use statistical tests to analyze the data and get a more quantitative sense.

4.     In study A the author has used 4 dogs, please mention the sex of the dogs. In study B, 8 females and 1 male dog were used. It was observed that sex-dependent differences in the PK of drugs in dogs. How can the author explain if there are any sex-dependent differences in the PK results?

5.     Were there sex-dependent differences in the toxicity?

6.     Why did the author decide to administer 17-AAG via the IV route instead of orally? Was there any bioavailability issue reported? If so please cite it.

7.     Line 232, instead of “these authors…..”, please write it to XXXX et al.

8.     The author reported that 3/9 dogs (~33%) of the study population showed elevation in ALT, AST, and GGT. It is also reported that some hypersensitivity reactions, Kupffer cell mobilization, and vacuolar degeneration in the tubular epithelium in the kidneys. However, the author wrote in line 288, that 17-AAG was found with minimal side and adverse effects. This is a contradictory statement to the observed results. Please, correct the statement based on the observed results. 

Comments on the Quality of English Language

None

Author Response

Reviewer #2

  1. The author can show the PK/PD graph to get a more quantitative sense of the data interpretation.
  1. As this is a high protein-bound compound, please use protein-adjusted IC50 in showing the PK/PD relationship.

Answer - We thank the referee for his/her valuable contributions. Unfortunately, we were unable to address both suggestions in this paper. A sentence has been added to the Discussion section to indicate that these evaluations should be addressed in future work.

OK Line 481-485: In future work, to better characterize the pharmacokinetic profile of 17-AAG, it should be advantageous provide a more quantitative data, involving PK/PD analysis and the presentation of the PK/PD relationship using a protein-adjusted IC50. 

  1. Throughout the manuscript, I didn’t see any statistical comparison of the results. Please, use statistical tests to analyze the data and get a more quantitative sense.

Answer: We acknowledge the referee's concerns regarding the statistical analyses of the treatment impact on several parameters evaluated in this study, which are addressed throughout the text. Additionally, a "Statistical Analysis" subsection has been added to the "Materials and Methods" section as follows:

OK Line 150-166: Significant differences in AUC and Cmax were observed when comparing doses of 50, 100, 150, and 200 mg/m², as well as between 50, 100, and 250 mg/m². However, no differences were found between doses of 150 or 200 mg/m² and 250 mg/m². We only found differences between half-life (t½) values between doses of 150 and 200 mg/m². Clearance of 17-AAG from plasma was, on average, 0.026 ± 0.010 L/h/m², with a coefficient of variation (CV) of 36%, while the CV between animals ranged from 1% to 20% that were not statistically different among doses. Similarly, no significant differences were observed among the various doses for distribution volume (DV) (Table 1). Figures 1 and Table 1 respectively illustrate that both AUC (measured over time) and Cmax increased proportionally with administered doses of 17-AAG, showing a linear correlation between doses ranging from 50 to 200 mg/m2. However, no correlation was observed between plasma concentration and a dose of 250 mg/m² (Figure 2). Also, we observed a rapid decline of 17-AAG in plasma regardless of the dose applied (Figure 2). 

OK Lines 280-288: Regarding the pharmacokinetic differences among dogs given three doses of 150 mg/m2 of 17-AAG, a significant difference was observed only in the t½ between the first and third administrations. No changes in Cmax values were observed in either plasma or interstitial fluid at any of the assessments undertaken after each 150 mg/m2 dose. Significant differences in AUC were observed when comparing the time intervals of 0-8 hours with 0-24 hours, and 0-6 hours with 0-24 hours. No significant differences were found among other intervals or across different administrations (Tables 3 and 4).

OK Lines 320-338: When considering hematological parameters and plasma biochemistry in animals after the administration of 17-AAG , only the total protein showed a significant difference among dogs in relation to drug concentration. Dogs receiving higher doses exhibited elevated levels of total protein in their serum. Plasma biochemistry analyses revealed that the enzymes, alanine aminotransferase (ALT) (28.57 ± 4.29 to 173.33 ± 49.56 U/L), aspartate aminotransferase (AST) (27.85 ± 3.80 to 248.20 ± 85.80 U/L), and Gamma-Glutamyl Transferase (GGT) (1.60 ± 0.06 to 12.70 ± 0.50 U/L) showed to be elevated in treated dogs but these differences revealed not to be statically distinct and reversible after drug washout.

When analyzing the frequency of side effects in dogs receiving varying doses of 17-AAG, increases in AST were noted in the plasma of dogs treated with 100 mg/m² (50% of animals), 150 mg/m² (50%), 200 mg/m² (100%), and 250 mg/m² (100%). However, these elevations were reversible. In animals where multiple doses of 17-AAG at 150 mg/m² were administered, elevated levels of ALT and AST were observed in only 33% of the dogs (3 out of 9). By the fourth day, elevated transaminase levels persisted in only one animal (Table 5).

OK Lines: 925-929 in Material and methods Section:

4.6. Statistical analysis

Pharmacokinetic parameters of 17-AAG were compared using a paired Student’s t-test. Hematological parameters and plasma biochemistry were evaluated with a two-way ANOVA, considering time points and doses.

  1. In study A the author has used 4 dogs, please mention the sex of the dogs. In study B, 8 females and 1 male dog were used. It was observed that sex-dependent differences in the PK of drugs in dogs. How can the author explain if there are any sex-dependent differences in the PK results? Were there sex-dependent differences in the toxicity?

Answer - We thank the referee for their concerns regarding the influence of sex on drug toxicity. We found no information in the literature about sex-dependent differences in the pharmacokinetics (PK) of 17-AAG treatment. The enrollment of healthy dogs is highly restricted by ethical considerations, and predominantly female animals were available for the study due to various factors, including owner consent, clinical status, and negative leishmaniasis tests. Therefore, we decided to conduct the study with these animals.

A short sentence clarifying this limitation has been included in the Discussion section: 

OK Line 530-537: One limitation of this study in assessing the safety and tolerability of 17-AAG is its predominant use of female dogs. Ethical considerations significantly constrain the enrollment of healthy dogs, limiting our capacity to recruit new animals for the research. Consequently, the study was conducted with the dogs that were initially recruited. Additionally, it is important to note that the literature lacks information on sex-dependent differences in the pharmacokinetics (PK) of 17-AAG treatment.

  1. Why did the author decide to administer 17-AAG via the IV route instead of orally? Was there any bioavailability issue reported? If so please cite it.

Answer: Thank you for the referee 's valuable comment. Please refer to answer 2 to referee 1 reproduced below.

OK Line 611-654: The major obstacle for delivery of 17-AAG related to its limited aqueous solubility (ca. 0.01 mg/mL) can encourage studies of special formulations for better solubilization and delivery of the drug, by various administration routes, including oral. Although oral administration seems to be a viable route for treatment, given that leishmaniasis is a systemic disease affecting multiple organs, the majority of literature involving 17-AAG, in both animal models and human studies, predominantly uses intravenous administration [54,9497]. According to the National Institute of Cancer's guide, "Turning Molecules into Medicines for Public Health https://dtp.cancer.gov/timeline/posters/AGG_Geldamycin.pdf," mice with MCF-7 breast tumors treated with various doses of 17-AAG exhibited a maximum plasma half-life of 4.4 hours following a 40 mg/kg intravenous dose, yet the drug was undetectable after oral administration. Biopharmaceutical strategies can be useful in this process, such as the incorporation of 17-AAG into micelles with special polymers, liposomes and other delivery systems, on nanometric scales. Xiong et al. [98] developed amphiphilic block copolymer (AB) micelles composed of degradable amphiphilic diblock polymers of poly(ethylene oxide)-block-poly(D,L-lactide) - PEO-b-PDLLA - as nanocarriers for solubilizing 17-AAG, and compared its pharmacokinetic behavior with a current formulation of 17-AAG in Cremophor EL (CrEL), ethanol (EtOH) and Polyethylene Glycol (PEG400) - CrEL-EtOH-PEG400. Katragadda et al. [99] developed micellar nanocarriers for concomitant delivery of paclitaxel and 17-allylamino-17-demethoxygeldanamycin (17-AAG) for cancer therapy by a solvent evaporation method. Our research group has already been studying formulation possibilities, using liposomal systems containing 17-AAG on Leishmania (L) amazonensis amastigotes. We showed promising results in the development of liposomes loaded with 17-AAG:HPβCD, facilitating drug solubilization that can subsequently enhance the distribution of the inhibitor systemically, both orally and intravenously. In fact, this nanoformulation, particularly when incorporated into HPβCD liposomes at 0.006 nM, achieved nearly complete clearance of L. amazonensis amastigotes inside macrophages after 48 hours of in vitro treatment. These findings underscore the potential of nanotechnology and drug delivery systems to enhance the antileishmanial efficacy and reduce the toxicity of 17-AAG. The results support evidence that nanotechnology and drug delivery systems could be used to increase the antileishmanial efficacy and potency of 17-AAG in vitro, while also resulting in reduced toxicity that indicates these formulations may represent a potential therapeutic strategy against leishmaniasis [100]. This aids in the administration of the medicine in a real-world context, improving adherence and enabling better monitoring of the treatment of dogs with CVL, and contributes to pharmacovigilance in studies involving the use of medicines in veterinary medicine.

In the Materials and Methods section, we have also included an explanation for choosing intravenous administration of the 17-AAG formulation as follows:

OK Lines 776-783: In both Studies A and B, intravenous administration of the formulations containing 17-AAG was chosen because the majority of studies in both animal models and human clinical trials, as cited in the literature, have utilized this administration route [51,54,95,96]. Control animals were the same dogs that, prior to treatment, received an intravenous injection of the placebo solution. After that, samples were collected for hematological and biochemical analyses described in Section 4.2 Animal selection and care procedures

  1. Line 232, instead of “these authors…..”, please write it to XXXX et al. 

Answer: We sincerely appreciate the reviewer's comments. In response to these comments, we have revised the text.

  1. The author reported that 3/9 dogs (~33%) of the study population showed elevation in ALT, AST, and GGT. It is also reported that some hypersensitivity reactions, Kupffer cell mobilization, and vacuolar degeneration in the tubular epithelium in the kidneys. However, the author wrote in line 288, that 17-AAG was found with minimal side and adverse effects. This is a contradictory statement to the observed results. Please, correct the statement based on the observed results. 

Answer: We sincerely appreciate the reviewer's comments. We apologize for the oversight and have revised the text accordingly.

Round 2

Reviewer 1 Report

Comments and Suggestions for Authors

The authors have addressed most of the comments and suggestions. Overall, the manuscript is well-written and presents important findings that contribute to the field of veterinary medicine, particularly in the treatment of CVL.

Comments on the Quality of English Language

The quality of the English language in the manuscript appears generally good based on the sections reviewed